# Risk of Dry Eye Syndrome in Patients with Orbital Fracture: A Nationwide Population-Based Cohort Study

**DOI:** 10.3390/healthcare9050605

**Published:** 2021-05-18

**Authors:** Cindy Yi-Yu Hsu, Junior Chun-Yu Tu, Chi-Hsiang Chung, Chien-An Sun, Wu-Chien Chien, Hsin-Ting Lin

**Affiliations:** 1Department of Ophthalmology, Mackay Memorial Hospital, Taipei 104, Taiwan; yiyu0610@gmail.com; 2Department of Plastic Surgery, Chang Gung Memorial Hospital, Chung Gung Medical University, Taoyuan 333, Taiwan; jr_tu@hotmail.com; 3School of Public Health, National Defense Medical Center, Taipei 114, Taiwan; g694810042@gmail.com; 4Department of Medical Research, Tri-Service General Hospital, National Defense Medical Center, Taipei 114, Taiwan; 5Taiwanese Injury Prevention and Safety Promotion Association (TIPSPA), Taipei 114, Taiwan; 6Department of Public Health, College of Medicine, Fu-Jen Catholic University, New Taipei City 242, Taiwan; 040866@mail.fju.edu.tw; 7Big Data Research Center, College of Medicine, Fu-Jen Catholic University, New Taipei City 242, Taiwan; 8Graduate Institute of Life Sciences, National Defense Medical Center, Taipei 114, Taiwan; 9Department of Ophthalmology, Tri-Service General Hospital, National Defense Medical Center, Taipei 114, Taiwan; 10Graduate Institute of Medical Sciences, National Defense Medical Center, Taipei 114, Taiwan

**Keywords:** dry eye syndrome, orbital fracture, ocular surface

## Abstract

This study aimed to investigate whether orbital fracture increases the risk of dry eye syndrome (DES) and identified the profile of prognostic factors. We studied a cohort from the Taiwan National Health Insurance Research Database (NHIRD). Overall, 46,179 and 184,716 participants were enrolled in the study and control groups, respectively. Each patient in the case group was age- and gender-matched to four individuals without orbital fracture that served as the control group. Cox proportional hazards analysis regression was used to estimate the risks of incident DES. During the follow-up period, the case group was more likely to develop incident DES (0.17%) than the control group (0.11%) (*p* = 0.001). Multivariate Cox regression analysis demonstrated that the case group had a 4.917-fold increased risk of DES compared to the controls. In the stratified age group, orbital fracture had the highest impact on patients aged 18–29 years. Furthermore, patients with orbital roof fracture have a greater risk of developing DES. Regardless of whether having received surgery or not, the patients with orbital fracture have higher risks of DES. Our study demonstrated that orbital fracture increases the risk of developing subsequent DES. Early recognition by thorough examinations with raised awareness in the clinical setting could preserve visual function and prevent further complications.

## 1. Introduction

The orbit is a bony cavity that contains the globe, extraocular muscles, nerves, fat, and blood vessels. Blunt trauma to the orbital rim leads to orbital fractures and causes damages to the surrounding facial bones and soft tissues [1,2,3]. Trauma to the eye and surrounding parts accounts for approximately 3% of all emergency department visits in the United States [4]. The predominant etiology of injury was violence (physical assault) followed by traffic accidents and sports injuries [5,6,7,8]. Males in their thirties are the most susceptible population [3,6,7,9]. The orbital fracture is often described according to the location of the injury, such as the floor, roof, medial wall, and lateral wall, with the orbital floor as the most common isolated orbital bone fracture site [6,8]. Additionally, a systemic review study found that 43 patients among 532 orbital fractures (8.1%) have led to a decrease in visual acuity [10].

To what extent the orbital trauma will affect ocular tear film stability is unclear [11]. Based on clinical experience, we have hypothesized that orbital fracture might be related to subsequent dry eye syndrome (DES). There has yet to be a large population study to support this hypothesis. Therefore, we conducted a longitudinal nationwide population-based cohort study using the Taiwan National Health Insurance Research Database (NHIRD).

DES is a multifactorial disease of the tears and the ocular surface that results in symptoms of discomfort, visual disturbance, and tear film instability with potential damage to the ocular surface. It is accompanied by increased osmolarity of the tear film and the inflammation of the ocular surface. A decrease in visual acuity associated with daily acts of gazing has been proven in dry eye patients [12]. The mechanism is related to disrupted tear film causing ocular surface irregularity [12,13,14]. DES increases with age and the prevalence is higher in women compared to men. The odds for DES increase 35% for each additional 10 years of age and the odds also increase for women [15]. Several independent risk factors have been found to be associated with DES: diabetes, connective tissue disease, hepatitis C, total to high-density lipoprotein cholesterol ratio, postmenopausal estrogen therapy, antihistamines, antidepressants, smoking status, caffeine use, contact lenses, and video display terminal exposure for more than 6 h/day [15,16]. Therefore, DES is considered a complex multi-factorial disease.

This study aimed to investigate whether orbital fracture increases the risk of DES. Moreover, potential risk factors, including several diseases and medications that may induce DES, were analyzed in the multivariable model. Furthermore, we discussed the association between surgery treatment for orbital fracture and DES.

## 2. Method

### 2.1. Data Resource and Ethics Declaration

The claims data used in the current study were accessed from the 2005 Longitudinal Health Insurance Database (LHID), which was derived from the NHIRD. The LHID was a subset of the NHIRD. It contained information from 2 million people and was used in the present study that randomly sampled individuals between 2000 and 2015. There was no significant difference in the distribution of sex, age, and insured premium between the LHID and the original NHIRD. Taiwan has initiated the National Health Insurance program in 1995. It covers approximately 99% of Taiwan’s population [17]. The data of LHID was randomly sampled from the NHIRD registry for the year 2005 by the database of the National Health Insurance Administration. The information available from the LHID include the demographic data of the subjects, their socioeconomic conditions, the residence of the subjects, the International Classification of Diseases-Ninth Revision (ICD-9), the International Classification of Diseases-Tenth Revision (ICD-10), and the medications used by each of the study subjects. The accuracy and high validity of diagnoses in the NHIRD have been demonstrated in previous articles [18]. The time interval of LHID ranges from 1 January 2000 to 31 December 2015, with a total study interval of about 15 years. This retrospective, population-based cohort study was approved by both the National Health Insurance Administration and the Institutional Review Board of Tri-Service General Hospital (TSGHIRB No. B-110-02). In addition, the need for informed consent was waived by the two institutions.

### 2.2. Study Participants

The flowchart of study sample selection from the LHID is shown in Figure 1. Of the total sample, 47,326 patients were followed up at the outpatient department more than 2 times with the diagnosis of orbital fracture or were hospitalized with orbital fracture being one of the, if not the only, diagnosis. The orbital fracture diagnostic codes include orbital floor closed fracture (ICD-9-CM code, 802.6), orbital floor open fracture (ICD-9-CM code, 8027), orbital roof closed fracture (ICD-9-CM code, 801.0~801.4), and orbital roof open fracture (ICD-9-CM code, 801.5~801.9). The index date was defined as the date of newly diagnosed orbital fracture. Furthermore, the following exclusion criteria were applied: being diagnosed with DES before the index date; being diagnosed with orbital fracture before the index date; having a diagnosis of severe ocular trauma at any time; having received eyeball removal surgery before the index date; without tracking; age < 18 years; and gender unknown. Finally, 46,179 patients matched our criteria and were assigned to the study cohort. For each orbital fracture patient, the four comparisons (1:4) were frequency matched by age (each 5-year span), sex, and index date as the control cohort.

### 2.3. Main Outcome Measurement

The development of DES was defined as the main outcome in the current study, which was based on the DES diagnosis (ICD-9-CM code, 375.15) after the index date. Moreover, only those patients having received a diagnosis of DES by an ophthalmologist were considered as having achieved the primary outcome and included in this study. In clinical practice, ICD-9/ICD-10 codes for “unspecific corneal disorder” may also be used for some forms of dry eye disease. However, these codes were eliminated to prevent overestimation and confusion of the primary outcome.

### 2.4. Demographic Variables and Comorbidities

To standardize the health condition of participants and to compare the baseline characteristics between the two groups, we considered the effects of demographic conditions including age, gender, urbanization level, income level, and the following comorbidities in the analysis: hypertension, diabetes mellitus, hyperlipidemia, ischemic heart diseases, congestive heart failure, chronic obstructive pulmonary disease, liver disease, and rheumatic disease. Moreover, in this study, we included several common factors of DES, such as connective tissue disease, multiple sclerosis (MS), osteoporosis, Bell’s palsy, and Parkinson’s disease to evaluate the confounding effects of orbital fracture. Additionally, to make the ocular condition of the study population more homogenous, we included the effect of trachoma, blepharitis, hordeolum, and glaucoma in the multivariable model. We then longitudinally followed the patients’ condition from the index date until the date of DES diagnosis or until the last date of data collection from the LHID, which is 31 December 2015.

### 2.5. Statistical Analysis

Statistical Product and Service Solutions (SPSS) 22nd edition (Armonk, NY, USA: IBM Corp.) was used for all the statistical analyses in the current study. The demographic features and common comorbidities between the orbital fracture patients and the control cohort were compared using the Chi-square test or Fisher’s exact test. The mean ages (continuous data) of both cohorts were measured using Student’s *t*-test. Then, the Cox proportional hazard regression was adopted to yield adjusted hazard ratios (aHR) of DES by incorporating the above demographic data, ocular diseases, and systemic comorbidities in the multivariable analysis. The incidence rate (per 105 person-years) of DES was calculated according to sex, age, and comorbidities for each cohort. For the subgroup analysis, the sensitivity analysis with aHR of DES that stratified by the surgery or not and orbital fracture types were conducted. In addition, the cumulative risks of DES were calculated with the Kaplan–Meier method and compared by log-rank test. Two-tailed *p*-values < 0.05 was considered as statistically significant.

## 3. Result

### 3.1. Sample Characteristics

The baseline demographic characteristics and common comorbidities of the case group and control group are shown in Table 1. The present study included 46,179 patients with newly diagnosed orbital fracture and 184,716 patients in the control cohort between 2000 and 2015. No significant differences were noted between the patients and controls in sex and age distribution at baseline. Of the patients with orbital fracture, 33,115 (71.71%) were males, and 15,295 (33.12%) were aged 18–29 years (mean age, 42.32 ± 17.99 years). The case group was more likely to have low insured premium, residence in higher urbanized areas, and high health care level at the time of the index date. Lower prevalence of most concomitant comorbidities (except Bell’s palsy) was noted in the case group. The prevalence of trachoma, blepharitis, and hordeolum had no significant difference between the two groups.

The distributions of incident DES and related clinical manifestations for the two groups during the 15-year follow-up are presented in Table 2. Compared with the control group (0.11%), the orbital fracture cohort had a higher incidence of DES (0.17%) during the follow-up period (*p* = 0.001). The average ages were 45.72 ± 19.04 and 49.11 ± 19.25 years for the study and the comparison cohort, respectively (*p* < 0.001). The mean follow-up time was 10.29 ± 15.95 years in the orbital fracture cohort and 10.75 ± 9.91 years in the control cohort (Appendix A). The mean duration to develop DES in the orbital fracture cohort was 4.13 ± 2.84 years, which is shorter than the control cohort (6.91 ± 4.57 years; Appendix A).

### 3.2. Kaplan–Meier Model for the Cumulative Risk of DES

A Kaplan–Meier graph of the cumulative risks of incident DES is shown in Figure 2, and the log-rank test revealed that the orbital fracture cohort had significantly higher cumulative risks than the control group (*p* < 0.001). The Kaplan–Meier analysis indicated that, in the third year, the incidence of DES was higher in the orbital fracture cohort than in the general population cohort (*p* = 0.036), a finding that persisted until the end of the follow-up (Appendix A).

### 3.3. Comparisons of the Prevalence and Risk of DES

Orbital fracture patients have a higher risk of DES compared with the control cohort (crude HR = 4.736 (95% CI, 3.622–6.193); *p* < 0.001; Table 3). After adjusting for sex, age, urbanization of residence areas, and other concomitant comorbidities, the adjusted HR was 4.917 (95% CI = 3.716–6.507; *p* < 0.001), indicating that patients with orbital fracture had a 4.917-fold increased risk of incident DES compared to controls. Notably, the risk of DES among the female patients with orbital fracture was significantly higher than male patients with orbital fracture, by a multiple of 1.523. Moreover, the adjusted risk of DES in the 18–29-year age group was 3.810-fold than the age group of ≥60 years (95% CI = 2.367–6.134; *p* < 0.001). Of the concomitant comorbidities, MS was the dominant factor for DES, with an adjusted HR of 8.064 (95% CI = 1.938–33.556; *p* = 0.004), followed by diffuse diseases of connective tissue (adjusted HR = 6.600 (95% CI, 3.999–10.894); *p* < 0.001) and blepharitis (adjusted HR = 6.246 (95% CI, 1.542–25.300); *p* = 0.010). Furthermore, orbital fracture patients with hordeolum, glaucoma, and Bell’s palsy have a greater likelihood of the developing DES. Significant factors of dry eye syndrome include orbital fracture, female, 18- to 29-year age group, MS, diffuse diseases of connective tissue, blepharitis, hordeolum, glaucoma, and Bell’s palsy.

### 3.4. Hazard Ratios Analysis of DES in the Patients with Orbital Fractures

Orbital fracture was associated with an increased risk of incident DES regardless of gender, age, season, and level of care (Table 4). The incidence rates of DES in the case group and the control group were 16.63 and 10.37 per 10^5^ person-years, respectively. Orbital fracture is the predominant factor among concomitant comorbidities.

### 3.5. Hazard Ratio Analysis of DES in the Patients Who Did and Did Not Receive Surgery of Orbital Fracture Subtypes

The patients with orbital fracture have a higher risk of DES whether they received surgery or not (Table 5). The patients who received surgery had a relatively higher risk of developing DES (adjusted HR = 1.097 [95% CI, 0.700–1.720]; *p* = 0.685) compared to patients without surgery. The Kaplan–Meier graph of the cumulative risks of incident DES showed the same result during the 15-year follow-up. Additionally, Table 5 shows the risks of different types of orbital fracture to DES. Patients with orbital roof fracture have a 1.566-fold risk of developing DES compare to orbital floor fracture patients (95% CI = 0.841–2.915; *p* = 0.157).

## 4. Discussion

The association between orbital fracture and DES remains unclear. Only one previous study pointed out a slightly lower amount of tear film in the affected eye compared with the unaffected eye among orbital floor fracture patients [11]. However, the study result did not meet the diagnosis criteria of DES. The study included only 23 participants, and the tear film quantity was measured by phenol red thread test. Furthermore, 10 of 23 patients (43%) had subnormal tear film values. Eight patients revealed a decreased tear production, while two displayed an excess in tear production (epiphora). Additionally, there was a considerably weak relationship between phenol red thread tests and symptoms of dry eyes [19]. Thus, a population-based research with large study number, confirmed diagnosis of DES, and multiple potential risk factors is warranted.

Using a population-based dataset, our study demonstrated that orbital fracture and DES were significantly associated even after adjusting for the patients’ demographic characteristics, comorbidities, and clinical pertinent covariates. During a 15-year follow-up, patients in the case group had an increased risk of DES, with an overall adjusted HR of 4.917 (95% CI, 3.716–6.507; *p* < 0.001), which was a 391.7% increase in the risk of developing DES compared to the controls. Notably, orbital roof fracture patients had relatively higher risk of developing DES compared to orbital floor fracture patients (adjusted HR = 1.566 (95% CI = 0.841–2.915); *p* = 0.157).

The possible mechanisms between the orbital fracture and DES are complex and may be associated with variable multi-factors. One plausible explanation for our observation was that the rate of DES occurrence following the orbital fracture was attributed to lacrimal gland injury. The lacrimal gland is located anteriorly in the superolateral aspect of the orbit, within the lacrimal fossa. The major source of tear fluid is from the lacrimal gland [20]. This suggests that the anatomical disruption and mechanical compression from orbital trauma contributed to the resulting injury of the lacrimal gland, including hematoma, edema, and vascular insufficiency. A previous study involved 200 cases of closed head injury admitted to a major teaching hospital. Ocular involvement was found in 167 (83.5%) cases, and two cases (1%) had lacrimal gland prolapse [21].

The inflammatory process during orbital fracture may activate the inflammatory reaction in the dry eye developing cycle. DES results from a combination of factors. One diagnostic classification scheme divides dry eye patients into those with aqueous tear deficiency and those with evaporative dry eye [22]. In aqueous tear deficiency, T-cell-mediated inflammation of the lacrimal gland occurs, which results in diminished secretion of the aqueous layer of the tear film and the propagation of inflammatory mediators on the ocular surface [23,24]. Therefore, severe orbital fracture, especially orbital roof fracture, may contribute to the development of DES, as demonstrated in the current study.

Isolated orbital roof fractures are uncommon. However, it is estimated that 1–9% of facial bone fractures involve the orbital roof [25,26]. In addition, orbital roof fractures are frequently associated with high-energy injuries to the head and face [26,27]. Several of these patients (13–19%) have multi-system injuries, most of which are neurologic (57–90%) [27,28,29]. These neurologic deficits could lead to higher risk of DES among orbital roof fracture patients.

In this study, we further evaluated the risks of DES associated with each comorbidity and studied the impact of orbital fracture on DES in association with these comorbidities (Table 3 and Table 4). Our results are consistent with those of previous epidemiological studies, which have revealed that female patients and several diseases are prone to DES, including osteoporosis, arthritis, connective tissue disease, and hyperlipidemia [15,16,30]. However, the tendency of DES increasing with age was not significant in our study. In contrast, orbital fracture patients aged 18–29 years had higher risks of DES than those aged ≥30 years. This might be due to the higher prevalence of orbital fracture in our cohort of patients in their thirties. Nevertheless, this does not neglect the fact that orbital fractures play a significant role in DES among the younger population.

A high prevalence of DES among patients with glaucoma and MS was observed in previous studies [31,32,33,34,35]. Our study further confirms the increase risk of DES among patients with glaucoma and MS. The results suggest that preventive therapy and current therapeutic efficacy should be considered.

However, the results should be interpreted within the context of the following limitations. First, although we had done our best to adjust for the influence of socioeconomic status, there were several confounding factors for orbital fracture and DES that we could not obtain from the NHIRD, such as contact lens use, smoking habits, caffeine use, alcohol consumption, nutrition, and video display terminal exposure, including the use of mobile and electronic devices. Second, patients diagnosed with orbital fracture and incident DES were identified based on the insurance claims data rather than real medical documents. The severity and laterality of the injuries were not available because of the lack of detailed clinical information in the ICD-9 coding system. Third, the retrospective nature of study design may reduce the homogeneity of the patient population, even after propensity score matching with multiple systemic diseases. Finally, we did not analyze the type of orbital fracture (i.e., fracture of medial wall of orbit, naso-orbital ethmoid bone, or lateral wall of the orbit) due to the limitation of the ICD-9 coding system. Despite these limitations, the strength of this study is that it reports longitudinal results on the association between orbital fracture and the risk of subsequent DES events in a nationwide, population-based cohort.

## 5. Conclusions

Our study clarified the relationship between orbital fracture and increased risk of developing subsequent DES. Early recognition by thorough examinations with increased awareness in the clinical setting could preserve visual function, eliminate ocular symptoms, and prevent further complications.

## Figures and Tables

**Figure 1 healthcare-09-00605-f001:**
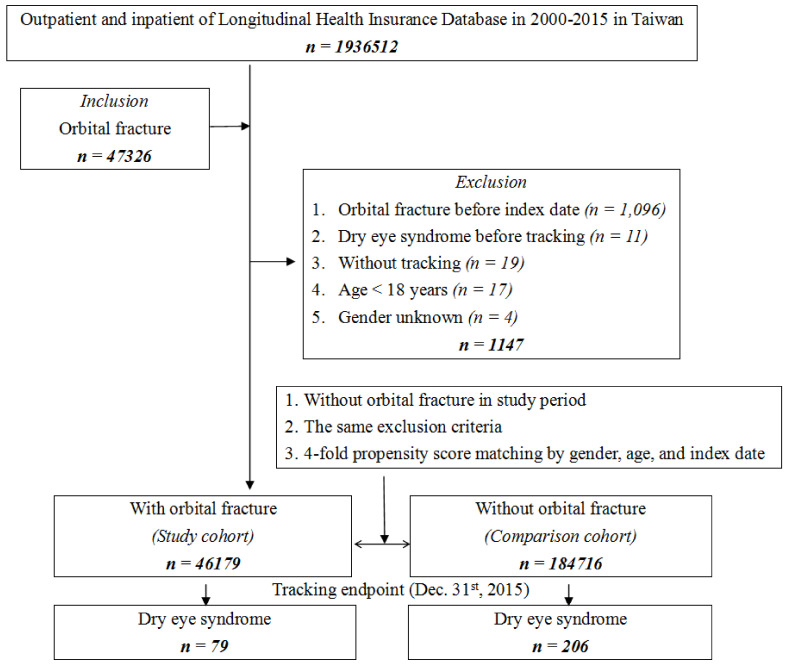
Flowchart of the study sample selection.

**Figure 2 healthcare-09-00605-f002:**
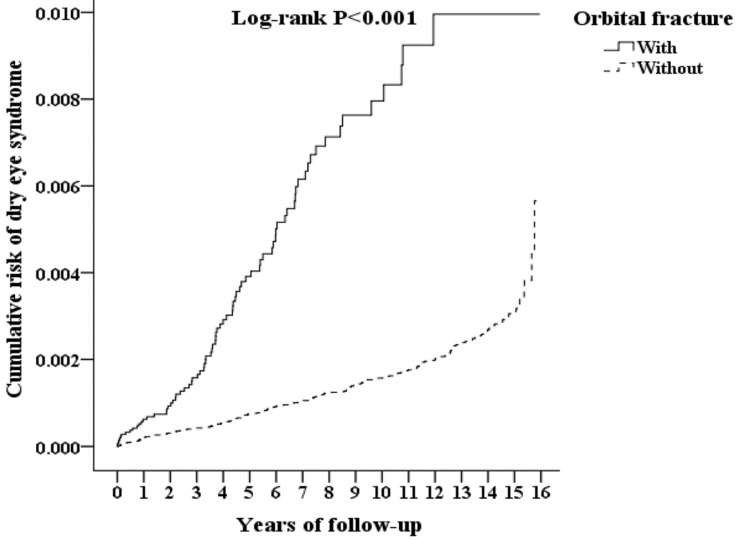
Kaplan–Meier for cumulative risk of dry eye syndrome aged 18 and over stratified by orbital fracture with log-rank test.

**Table 1 healthcare-09-00605-t001:** Characteristics of study in the baseline.

Orbital Fracture Variables	With		Without		*p*
	*n*	%	*n*	%	
Total	46,179	20.00	184,716	80.00	
Gender					0.999
Male	33,115	71.71	132,460	71.71	
Female	13,064	28.29	52,256	28.29	
Age (years)	42.32 ± 17.99		42.38 ± 17.81		0.518
Age group (yrs)					0.999
18–29	15,295	33.12	61,180	33.12	
30–39	7962	17.24	31,848	17.24	
40–49	7511	16.26	30,044	16.26	
50–59	6470	14.01	25,880	14.01	
≥60	8941	19.36	35,764	19.36	
Insured premium (NT$)					<0.001
<18,000	45,279	98.05	180,980	97.98	
18,000–34,999	699	1.51	2606	1.41	
≥35,000	201	0.44	1130	0.61	
Education levels (yrs)					0.337
<12	23,792	51.52	95,629	51.77	
≥12	22,387	48.48	89,087	48.23	
Diabetes mellitus (DM)	5862	12.69	33,957	18.38	<0.001
Hyperlipidemia	1562	3.38	16,416	8.89	<0.001
Hypertension (HTN)	8487	18.38	49,378	26.73	<0.001
Chronic kidney disease (CKD)	1900	4.11	20,174	10.92	<0.001
Coronary artery disease (CAD)	3428	7.42	32,220	17.44	<0.001
Congestive heart failure (CHF)	1293	2.80	13,204	7.15	<0.001
Stroke	4408	9.55	25,696	13.91	<0.001
Chronic obstructive pulmonary disease (COPD)	2803	6.07	24,213	13.11	<0.001
Chronic liver disease (CLD)	2847	6.17	29,364	15.90	<0.001
Osteoporosis	239	0.52	3065	1.66	<0.001
Rheumatoid arthritis (RA)	104	0.23	986	0.53	<0.001
Connective tissue disease (CTD)	98	0.21	807	0.44	<0.001
Sarcoidosis	4	0.01	30	0.02	0.287
Trachoma	74	0.16	242	0.13	0.137
Multiple sclerosis (MS)	8	0.02	90	0.05	0.004
Bell’s palsy	896	1.94	569	0.31	<0.001
Parkinson disease	468	1.01	2900	1.57	<0.001
Blepharitis	17	0.04	109	0.06	0.079
Hordeolum	29	0.06	177	0.10	0.039
Glaucoma	254	0.55	1354	0.73	<0.001
CCI_R	0.04 ± 0.39		0.31 ± 1.34		<0.001
Anti-HTN drugs	7923	17.16	41,356	22.39	<0.001
Antidepressants drugs	2656	5.75	9752	5.28	<0.001
Anti-CA drugs	136	0.29	555	0.30	0.834
Anti-Parkinson drugs	302	0.65	1915	1.04	<0.001
Anti-ulcer drugs	131	0.28	228	0.12	<0.001
Muscle spasm drugs	206	0.45	701	0.38	0.041
Decongestant drugs	389	0.84	1315	0.71	0.003
Antihistamines drugs	1136	2.46	4499	2.44	0.762
Anesthetics drugs	117	0.25	430	0.23	0.416
Season					0.999
Spring (Mar–May)	11,178	24.21	44,712	24.21	
Summer (Jun–Aug)	11,339	24.55	45,356	24.55	
Autumn (Sep–Nov)	12,069	26.14	48,276	26.14	
Winter (Dec–Feb)	11,593	25.10	46,372	25.10	
Location					<0.001
Northern Taiwan	15,789	34.19	74,088	40.11	
Middle Taiwan	12,434	26.93	51,859	28.07	
Southern Taiwan	14,925	32.32	47,312	25.61	
Eastern Taiwan	2865	6.20	10,583	5.73	
Outlets islands	166	0.36	874	0.47	
Urbanization level					<0.001
1 (The highest)	16,440	35.60	63,131	34.18	
2	19,802	42.88	76,262	41.29	
3	3440	7.45	16,404	8.88	
4 (The lowest)	6497	14.07	28,919	15.66	
Level of care					<0.001
Hospital center	19,913	43.12	54,086	29.28	
Regional hospital	20,770	44.98	55,894	30.26	
Local hospital	5496	11.90	74,736	40.46	

*p*: Chi-square/Fisher exact test on category variables and *t*-test on continue variables.

**Table 2 healthcare-09-00605-t002:** Characteristics of study in the endpoint.

Orbital Fracture	With		Without		*p*
Variables	*n*	%	*n*	%	
Total	46,179	20	184,716	80	
Dry eye syndrome					0.001
Without	46,100	99.83	184,510	99.89	
With	79	0.17	206	0.11	
Gender					0.999
Male	33,115	71.71	132,460	71.71	
Female	13,064	28.29	52,256	28.29	
Age (years)	45.72 ± 19.04		49.11 ± 19.25		<0.001
Age group (yrs)					<0.001
18–29	12,414	26.88	35,791	19.38	
30–39	8152	17.65	37,077	20.07	
40–49	7360	15.94	30,221	16.36	
50–59	6801	14.73	25,180	13.63	
≥60	11,452	24.8	56,447	30.56	
Insured premium (NT$)					<0.001
<18,000	45,279	98.05	180,980	97.98	
18,000–34,999	699	1.51	2606	1.41	
≥35,000	201	0.44	1130	0.61	
Education levels (yrs)					
<12	46,179	100	184,716	100	
≥12		0		0	
DM	5862	12.69	33,957	18.38	<0.001
Hyperlipidemia	1562	3.38	16,416	8.89	<0.001
HTN	8487	18.38	49,378	26.73	<0.001
CKD	1900	4.11	20,174	10.92	<0.001
CAD	3428	7.42	32,220	17.44	<0.001
CHF	1293	2.8	13,204	7.15	<0.001
Stroke	4408	9.55	25,696	13.91	<0.001
COPD	2803	6.07	24,213	13.11	<0.001
CLD	2847	6.17	29,364	15.9	<0.001
Osteoporosis	239	0.52	3065	1.66	<0.001
RA	104	0.23	986	0.53	<0.001
CTD	98	0.21	807	0.44	<0.001
Sarcoidosis	4	0.01	30	0.02	0.287
Trachoma	74	0.16	242	0.13	0.137
MS	8	0.02	90	0.05	0.004
Bell’s palsy	896	1.94	569	0.31	<0.001
Parkinson disease	468	1.01	2900	1.57	<0.001
Blepharitis	17	0.04	109	0.06	0.079
Hordeolum	29	0.06	177	0.1	0.039
Glaucoma	254	0.55	1354	0.73	<0.001
CCI_R	0.04 ± 0.39		0.31 ± 1.34		<0.001
Anti-HTN drugs	7923	17.16	41,356	22.39	<0.001
Antidepressants drugs	2656	5.75	9752	5.28	<0.001
Anti-CA drugs	136	0.29	555	0.3	0.834
Anti-Parkinson drugs	302	0.65	1915	1.04	<0.001
Anti-ulcer drugs	131	0.28	228	0.12	<0.001
Muscle spasm drugs	206	0.45	701	0.38	0.041
Decongestant drugs	389	0.84	1315	0.71	0.003
Antihistamines drugs	1136	2.46	4499	2.44	0.762
Anesthetics drugs	117	0.25	430	0.23	0.416
Season					<0.001
Spring	10,654	23.07	45,045	24.39	
Summer	11,535	24.98	48,254	26.12	
Autumn	12,846	27.82	47,521	25.73	
Winter	11,144	24.13	43,896	23.76	
Location					<0.001
Northern Taiwan	16,139	34.95	73,831	39.97	
Middle Taiwan	12,431	26.92	52,105	28.21	
Southern Taiwan	14,703	31.84	47,128	25.51	
Eastern Taiwan	2740	5.93	10,798	5.85	
Outlets islands	166	0.36	854	0.46	
Urbanization level					<0.001
1 (The highest)	15,777	34.16	60,659	32.84	
2	19,613	42.47	78,985	42.76	
3	3537	7.66	15,702	8.5	
4 (The lowest)	7252	15.7	29,370	15.9	
Level of care					<0.001
Hospital center	18,198	39.41	60,196	32.59	
Regional hospital	20,537	44.47	70,919	38.39	
Local hospital	61,045	26.44	7444	16.12		

*p:* Chi-square/Fisher’s exact test on category variables and *t*-test on continue variables. Adjusted HR: multivariable analysis included sex, age, covariates, and comorbidities (hypertension, diabetes mellitus, hyperlipidemia, ischemic heart diseases, congestive heart failure, chronic obstructive pulmonary disease, liver disease, rheumatic disease, connective tissue disease, multiple sclerosis, osteoporosis, Bell’s palsy, Parkinson’s disease, trachoma, blepharitis, hordeolum, and glaucoma) and medications (listed in Table 1).

**Table 3 healthcare-09-00605-t003:** Factors of dry eye syndrome by using Cox regression.

Variables	Crude HR	95% CI	95% CI	*p*	Adjusted HR	95% CI	95% CI	*p*
Orbital fracture (Reference: without)	4.736	3.622	6.193	<0.001	4.917	3.716	6.507	<0.001
Male (Reference: Female)	1.614	1.273	2.046	<0.001	1.523	1.187	1.954	
Age (Reference: ≥60)								
18–29	2.332	1.539	3.533	<0.001	3.810	2.367	6.134	<0.001
30–39	0.745	0.518	1.071	0.111	1.348	0.89	2.044	0.159
40–49	0.869	0.615	1.228	0.427	1.400	0.953	2.055	0.086
50–59	1.14	0.826	1.574	0.424	1.512	1.07	2.136	0.019
Insured premium (Reference: <18,000)								
18,000–34,999	0.432	0.107	1.735	0.236	0.448	0.111	1.801	0.258
≥35,000	2.755	1.027	7.392	0.044	3.375	1.252	9.101	0.016
Education levels (years) (Reference: <12)								
Comorbidities (Reference: Without)								
DM	1.354	1.059	1.730	0.016	1.243	0.936	1.650	0.133
Hyperlipidemia	1.619	1.223	2.144	0.001	1.472	1.08	2.006	0.014
HTN	1.350	1.068	1.705	0.012	1.080	0.801	1.456	0.613
CKD	1.003	0.718	1.401	0.988	1.007	0.514	1.358	0.098
CAD	1.448	1.132	1.853	0.003	1.269	0.943	1.708	0.115
CHF	1.203	0.840	1.721	0.313	1.006	0.609	1.348	0.626
Stroke	1.348	1.032	1.761	0.029	1.162	0.86	1.570	0.328
COPD	1.741	1.343	2.257	<0.001	1.836	1.375	2.452	<0.001
CLD	1.126	0.854	1.486	0.399	1.172	0.875	1.570	0.288
Osteoporosis	2.801	1.759	4.462	<0.001	1.993	1.211	3.280	0.007
RA	5.576	3.195	9.730	<0.001	2.577	1.383	4.800	0.003
CTD	11.640	7.530	17.993	<0.001	6.600	3.999	10.894	<0.001
Sarcoidosis	0.000	-	-	0.84	0.000	-	-	0.979
Trachoma	0.000	-	-	0.577	0.000	-	-	0.955
MS	11.569	2.879	46.492	0.001	8.064	1.938	33.556	0.004
Bell’s palsy	3.719	1.756	7.874	0.001	2.192	1.028	4.674	0.042
Parkinson disease	1.959	1.145	3.353	0.014	1.778	1.026	3.081	0.04
Blepharitis	7.655	1.905	30.763	0.004	6.246	1.542	25.300	0.01
Hordeolum	8.100	3.018	21.740	<0.001	5.734	2.055	16.003	0.001
Glaucoma	6.067	3.808	9.666	<0.001	4.960	3.071	8.009	<0.001
CCI_R	1.008	0.838	1.142	0.782	1.050	0.912	1.208	0.496
Medications (Reference: Without)								
Anti-HTN drugs	1.453	0.88	1.979	0.546	1.210	0.797	1.871	0.514
Antidepressants drugs	1.103	0.794	1.29	0.464	1.092	0.722	1.238	0.478
Anti-CA drugs	0.981	0.357	1.599	0.785	0.946	0.332	1.524	0.762
Anti-Parkinson drugs	1.584	0.877	2.601	0.578	1.423	0.758	2.34	0.588
Anti-ulcer drugs		0.300	2.990	0.876	1.678	0.245	2.593	0.835
Muscle spasm drugs	1.121	0.599	1.867	0.351	1.104	0.532	1.82	0.333
Decongestant drugs	1.266	0.451	2.384	0.623	1.298	0.489	2.415	0.601
Antihistamines drugs	1.986	0.230	4.350	0.927	1.834	0.202	4.030	0.911
Anesthetics drugs	0.989	0.149	2.846	0.933	1.006	0.164	3.000	0.976
Season (Reference: Spring)								
Summer	0.781	0.566	1.079	0.134	0.752	0.544	1.039	0.084
Autumn	0.686	0.495	0.951	0.024	0.641	0.462	0.890	0.008
Winter	0.841	0.609	1.163	0.296	0.828	0.599	1.145	0.254
Location (Reference: Northern Taiwan)								
Middle Taiwan	0.671	0.494	0.913	0.011	Multicollinearity with urbanization level			
Southern Taiwan	0.890	0.667	1.188	0.429	Multicollinearity with urbanization level			
Eastern Taiwan	1.324	0.882	1.987	0.176	Multicollinearity with urbanization level			
Outlets islands	0	-	-	0.912	Multicollinearity with urbanization level			
Urbanization level (Reference: 4)								
1 (The highest)	1.564	1.088	2.248	0.016	1.181	0.801	1.740	0.401
2	1.501	1.024	2.202	0.037	1.026	0.587	1.796	0.927
3	1.071	0.616	1.863	0.807	1.011	0.655	1.560	0.960
Level of care (Reference: Local hospital)								
Hospital center	2.537	1.753	3.671	<0.001	2.776	1.830	4.213	<0.001
Regional hospital	1.646	1.134	2.390	0.009	1.723	1.178	2.521	0.005

HR, hazard ratio; CI, confidence interval; Adjusted HR, Adjusted variables listed in the table. Adjusted HR: multivariable analysis included sex, age, covariates, and comorbidities (hypertension, diabetes mellitus, hyperlipidemia, ischemic heart diseases, congestive heart failure, chronic obstructive pulmonary disease, liver disease, rheumatic disease, connective tissue disease, multiple sclerosis, osteoporosis, Bell’s palsy, Parkinson’s disease, trachoma, blepharitis, hordeolum, and glaucoma) and medications (listed in Table 1).

**Table 4 healthcare-09-00605-t004:** Factors of dry eye syndrome stratified by variables listed in the table using Cox regression.

Orbital Fracture Stratified	With			Without (Reference)			With vs. Without (Reference)			
	Events	PYs	Rate (per 10^5^ PYs)	Events	PYs	Rate (per 10^5^ PYs)	Adjusted HR	95% CI	95% CI	*p*
Total	79	475,088.93	16.63	206	1,985,645.35	10.37	4.917	3.716	6.507	<0.001
Gender										
Male	44	335,656.21	13.11	118	1,409,299.03	8.37	4.803	3.629	6.356	<0.001
Female	35	139,432.71	25.10	88	576,346.32	15.27	5.043	3.811	6.674	<0.001
Age (yrs)										
18–29	19	55,747.48	34.08	16	90,383.93	17.70	5.906	4.463	7.816	<0.001
30–39	11	92,878.70	11.84	28	376,713.68	7.43	4.888	3.694	6.468	<0.001
40–49	11	79,470.94	13.84	30	355,420.38	8.44	5.031	3.802	6.657	<0.001
50–59	18	80,902.99	22.25	40	330,681.83	12.10	5.643	4.264	7.467	<0.001
≥60	20	166,088.82	12.04	92	832,445.53	11.05	3.343	2.526	4.423	<0.001
Insured premium (NT$)										
<18,000	77	465,576.68	16.54	202	1,943,410.70	10.39	4.881	3.689	6.459	<0.001
18,000–34,999	0	7798.00	0.00	2	31,835.37	6.28	0.000	-	-	0.979
≥35,000	2	1714.24	116.67	2	10,399.29	19.23	18.610	14.064	24.627	<0.001
Education levels (yrs)										
<12	46	248,072.48	18.54	138	1,045,925.08	13.19	4.311	3.258	5.705	<0.001
≥12	33	227,016.45	14.54	68	939,720.27	7.24	6.163	4.657	8.155	<0.001
DM										
Without	49	362,152.29	13.53	128	1,434,066.93	8.93	4.650	3.514	6.154	<0.001
With	30	112,936.64	26.56	78	551,578.42	14.14	5.763	4.355	7.626	<0.001
Hyperlipidemia										
Without	62	431,204.03	14.38	152	1,687,032.31	9.01	4.896	3.700	6.478	<0.001
With	17	43,884.89	38.74	54	298,613.04	18.08	6.572	4.966	8.696	<0.001
HTN										
Without	54	305,101.23	17.70	129	1,147,122.95	11.25	4.828	3.649	6.389	<0.001
With	25	169,987.70	14.71	77	838,522.40	9.18	4.913	3.713	6.502	<0.001
CKD										
Without	69	433,018.39	15.93	172	1,700,934.97	10.11	4.834	3.653	6.397	<0.001
With	10	42,070.54	23.77	34	284,710.38	11.94	6.106	4.614	8.080	<0.001
CAD										
Without	59	391,342.67	15.08	137	1,463,436.21	9.36	4.940	3.733	6.538	<0.001
With	20	83,746.26	23.88	69	522,209.14	13.21	5.545	4.190	7.337	<0.001
CHF										
Without	71	444,230.54	15.98	177	1,781,171.96	9.94	4.934	3.729	6.529	<0.001
With	8	30,858.39	25.92	29	204,473.39	14.18	5.608	4.238	7.420	<0.001
Stroke										
Without	57	381,249.02	14.95	150	1,591,558.41	9.42	4.866	3.678	6.440	<0.001
With	22	93,839.91	23.44	56	394,086.94	14.21	5.061	3.825	6.697	<0.001
COPD										
Without	59	410,992.47	14.36	142	1,623,732.70	8.75	5.036	3.805	6.664	<0.001
With	20	64,096.46	31.20	64	361,912.64	17.68	5.413	4.091	7.163	<0.001
CLD										
Without	59	404,963.89	14.57	153	1,568,057.99	9.76	4.581	3.462	6.061	<0.001
With	20	70,125.04	28.52	53	417,587.35	12.69	6.894	5.209	9.122	<0.001
Osteoporosis										
Without	73	468,938.39	15.57	188	1,934,814.70	9.72	4.915	3.714	6.504	<0.001
With	6	6150.54	97.55	18	50,830.65	35.41	8.451	6.386	11.183	<0.001
RA										
Without	77	472,765.82	16.29	198	1,968,579.60	10.06	4.968	3.754	6.574	<0.001
With	2	2323.11	86.09	8	17,065.75	46.88	5.634	4.258	7.455	<0.001
CTD										
Without	73	472,713.12	15.44	191	1,971,480.28	9.69	4.890	3.695	6.471	<0.001
With	6	2375.81	252.55	15	14,165.06	105.89	7.316	5.529	9.681	<0.001
Sarcoidosis										
Without	79	474,923.46	16.63	204	1,985,048.07	10.28	4.965	3.752	6.571	<0.001
With	0	165.47	0.00	2	597.28	334.85	0.000	-	-	0.976
Trachoma										
Without	79	472,879.08	16.71	205	1,981,210.18	10.35	4.953	3.743	6.554	<0.001
With	0	2209.85	0.00	1	4435.17	22.55	0.000	-	-	0.989
MS										
Without	78	474,990.41	16.42	205	1,984,386.50	10.33	4.876	3.685	6.453	<0.001
With	1	98.52	1015.01	1	1258.85	79.44	39.198	29.622	51.870	<0.001
Bell’s palsy										
Without	75	462,955.91	16.20	204	1,975,921.80	10.32	4.814	3.638	6.370	<0.001
With	4	12,133.02	32.97	2	9723.55	20.57	4.917	3.716	6.507	<0.001
Parkinson disease										
Without	76	464,221.80	16.37	199	1,935,404.09	10.28	4.885	3.691	6.464	<0.001
With	3	10,867.13	27.61	7	50,241.26	13.93	6.078	4.593	8.043	<0.001
Blepharitis										
Without	76	474,816.42	16.01	204	1,983,792.45	10.28	4.775	3.608	6.319	<0.001
With	3	272.51	1100.88	2	1852.90	107.94	31.288	23.644	41.403	<0.001
Hordeolum										
Without	76	474,256.27	16.03	202	1,982,298.80	10.19	4.824	3.646	6.384	<0.001
With	3	832.66	360.29	4	3346.55	119.53	9.247	6.988	12.237	<0.001
Glaucoma										
Without	76	470,302.16	16.16	198	1,962,759.09	10.09	4.914	3.714	6.503	<0.001
With	3	4786.77	62.67	8	22,886.26	34.96	5.500	4.156	7.278	<0.001
Anti-HTN drugs										
Without	56	393,561.26	14.23	139	1,541,068.35	9.02	4.840	3.657	6.404	<0.001
With	23	81,527.67	28.21	67	444,577.00	15.07	5.743	4.340	7.599	<0.001
Antidepressants drugs										
Without	50	447,758.69	11.17	144	1,880,811.35	7.66	4.474	3.381	5.921	<0.001
With	29	27,330.24	106.11	62	104,834.00	59.14	5.504	4.159	7.283	<0.001
Anti-CA drugs										
Without	66	473,689.49	13.93	166	1,979,679.10	8.39	5.097	3.852	6.745	<0.001
With	13	1399.44	928.94	40	5966.25	670.44	4.251	3.212	5.625	<0.001
Anti-Parkinson drugs										
Without	53	471,981.35	11.23	127	1,965,059.10	6.46	5.330	4.028	7.053	<0.001
With	26	3107.58	836.66	79	20,586.25	383.75	6.688	5.054	8.851	<0.001
Anti-ulcer drugs										
Without	61	473,740.94	12.88	187	1,983,194.35	9.43	4.189	3.166	5.544	<0.001
With	18	1347.99	1335.32	19	2451.00	775.19	5.284	3.993	6.993	<0.001
Muscle spasm drugs										
Without	70	472,969.19	14.80	188	1,978,109.60	9.50	4.777	3.610	6.322	<0.001
With	9	2119.74	424.58	18	7535.75	238.86	5.453	4.121	7.216	<0.001
Decongestant drugs										
Without	61	471,086.12	12.95	171	1,971,509.10	8.67	4.580	3.461	6.060	<0.001
With	18	4002.81	449.68	35	14,136.25	247.59	5.572	4.211	7.373	<0.001
Antihistamines drugs										
Without	65	463,399.49	14.03	173	1,937,281.10	8.93	4.819	3.641	6.376	<0.001
With	14	11,689.44	119.77	33	48,364.25	68.23	5.385	4.069	7.126	<0.001
Anesthetics drugs										
Without	59	473,885.00	12.45	164	1,981,022.85	8.28	4.614	3.486	6.105	<0.001
With	20	1203.93	1661.23	42	4622.50	908.60	5.609	4.239	7.422	<0.001
Season										
Spring	25	105,038.76	23.80	56	462,940.42	12.10	6.036	4.561	7.987	<0.001
Summer	18	121,136.73	14.86	51	508,305.36	10.03	4.543	3.433	6.012	<0.001
Autumn	13	137,573.52	9.45	48	548,918.18	8.74	3.315	2.505	4.387	<0.001
Winter	23	111,339.92	20.66	51	465,481.39	10.96	5.784	4.371	7.654	<0.001
Urbanization level										
1 (The highest)	29	135,162.15	21.46	55	589,023.47	9.34	7.049	5.327	9.328	<0.001
2	21	203,152.89	10.34	53	868,522.96	6.10	5.197	3.927	6.877	<0.001
3	15	44,460.22	33.74	44	171,456.22	25.66	4.033	3.048	5.337	<0.001
4 (The lowest)	14	92,313.67	15.17	54	356,642.70	15.14	3.073	2.322	4.066	<0.001
Level of care										
Hospital center	35	140,015.22	25.00	80	637,196.38	12.55	6.108	4.616	8.083	<0.001
Regional hospital	29	224,241.18	12.93	75	899,300.83	8.34	4.757	3.595	6.295	<0.001
Local hospital	15	110,832.53	13.53	51	449,148.13	11.35	3.656	2.763	4.839	<0.001

PYs, person-years; Adjusted HR, adjusted hazard ratio, adjusted for the variables listed in Table 3; CI, confidence interval. Adjusted HR: multivariable analysis included sex, age, covariates, and comorbidities (hypertension, diabetes mellitus, hyperlipidemia, ischemic heart diseases, congestive heart failure, chronic obstructive pulmonary disease, liver disease, rheumatic disease, connective tissue disease, multiple sclerosis, osteoporosis, Bell’s palsy, Parkinson’s disease, trachoma, blepharitis, hordeolum, and glaucoma) and medications (listed in Table 1).

**Table 5 healthcare-09-00605-t005:** Factors of dry eye syndrome with/without surgery of orbital fracture subtypes using Cox regression.

	Subgroup	Populations	Events	PYs	Rate (per 10^5^ PYs)	Adjusted HR	95% CI	95% CI	*p*	Adjusted HR	95% CI	95% CI	*p*
	Without orbital fracture	184,716	206	1,985,645.35	10.37	Reference							
	With orbital fracture	46,179	79	475,088.93	16.63	4.917	3.716	6.507	<0.001				
Surgery	Without surgery	23,007	37	236,003.86	15.68	4.718	3.276	6.796	<0.001	Reference			
	With surgery	23,172	42	239,085.07	17.57	5.113	3.596	7.269	<0.001	1.097	0.700	1.720	0.685
Orbital fracture subtypes	Orbital floor fracture	10,165	12	102,987.71	11.65	3.034	1.668	5.519	<0.001	Reference			
	Orbital roof fracture	36,014	67	372,101.21	18.01	5.534	4.114	7.443	<0.001	1.566	0.841	2.915	0.157
Surgery × Orbital fracture subtypes	Orbital floor fracture, without surgery	2125	1	20,540.24	4.87	1.431	0.200	10.238	0.721	Reference			
	Orbital roof fracture, without surgery	20,882	36	215,463.62	16.71	5.073	3.507	7.337	<0.001	3.423	0.468	25.049	0.226
	Orbital floor fracture, with surgery	8040	11	82,447.49	13.34	3.401	1.820	6.356	<0.001	2.664	0.342	20.733	0.349
	Orbital roof fracture, with surgery	15,132	31	156,637.59	19.79	6.209	4.175	9.234	<0.001	3.977	0.542	29.276	0.175

PYs, person-years; Adjusted HR, adjusted hazard ratio, adjusted for the variables listed in Table 3; CI, confidence interval.

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
