# Peer review of "Risk of Dry Eye Syndrome in Patients with Orbital Fracture: A Nationwide Population-Based Cohort Study"

_healthcare, 2021, doi:10.3390/healthcare9050605_

Round 1

Reviewer 1 Report

In this study titled “Risk of dry eye syndrome in patients with orbital fracture: A nationwide population-based cohort study”, the authors evaluate the risk of dry eye syndrome diagnosed with orbital fracture and the prevalence on patients with others DES factors. Overall, the manuscript is overwhelming and the results are hard to follow up along the manuscript. The results section is disorganized with large amount of data on tables with information that are irrelevant or not considered further. It is suggested to limit only to relevant data and include more graphics such as bars, pie chart, etc. for better visualization. The discussion and conclusion should be improved. Please find below some specific comments that must be addressed before considering for publication:

  1. The authors include several factors of DES including autoimmune diseases. Is there any data about Sjogren disease to include since is a common cause of DED?
  2. Split the results sections and add a subtitle highlighting the results that want to show.
  3. Can you please explain what is the clinical outcomes for DES diagnosis after orbital fracture? Is due to tear secretion deficiently? It is not clear in the method section what was the criteria for DES diagnosis, please explain what is ICD-9-CM code: 375.15. More importantly, was there any correlation of patients that develop DES with any other sign of eye damage or any other ocular disease? This information could indicate the DED causes such as nerve or vascular damage, anatomical changes, inflammation, etc.
  4. The factors of DES listed table 3 were diagnose before or after orbital fracture?
  5. Did the authors find increased of other eye diseases after orbital fracture? It would be important to determine the correlation between orbital fracture and the risk of other eye diseases.
  6. On pg 15. Please indicate which figure is referring these results. Please include only the data with the factors that have significant impact.
  7. Please elaborate the importance of this findings in the discussion section. What kind of preventive therapy or therapeutic treatment can be considered?
  8. The limitation of this study mentioned at the end of the discussion section should be mentioned at the beginning or in the method section prior to the interpretation of the results.
  9. Overall, English grammar should be revised. Eg. Section 2.4 “having received a diagnosis of DES diagnosis”
  10. Check the fonts. Eg second paragraph on the introduction and third paragraph on the discussion section.

Author Response

  1. The authors include several factors of DES including autoimmune diseases. Is there any data about Sjogren disease to include since is a common cause of DED?

Re:   Sjogren’s disease, also known as Sicca syndrome, is filed under diffuse diseases of the connective tissue. The ICD-9-CM code is 710

  1. Split the results sections and add a subtitle highlighting the results that want to show.

Re: Thank you for this suggestion. After revision, subtitles have now been added. The subtitles are as follows: 

3.1. Sample characteristics

3.2. Kaplan-Meier model for the cumulative risk of DES

3.3. Comparisons of the prevalence and risk of DES

3.4. Hazard ratios analysis of DES in the patients with orbital fractures

3.5. Hazard ratio analysis of DES in the patients who did and did not receive surgery of orbital fracture subtypes

  1. Can you please explain what is the clinical outcomes for DES diagnosis after orbital fracture? Is due to tear secretion deficiently? It is not clear in the method section what was the criteria for DES diagnosis, please explain what is ICD-9-CM code: 375.15. More importantly, was there any correlation of patients that develop DES with any other sign of eye damage or any other ocular disease? This information could indicate the DED causes such as nerve or vascular damage, anatomical changes, inflammation, etc.

Re: In this study, we examined a cohort from the Taiwan National Health Insurance database (NHIRD). This cohort is made up of 46,179 patients diagnosed by board-certified ophthalmologists across the nation. 

As there is no ICD-9 diagnosis code for DES itself, the most common ICD-9-CM Diagnosis Code used is 375.15, which is the code for Tear film insufficiency. 

The purpose of this study focuses on the correlation of DES with orbital wall fracture. Even though there might be some relation of DES with other signs of eye damage or other ocular disease, this was not discussed in this study. 

  1. The factors of DES listed table 3 were diagnose before or after orbital fracture?

Re: Table 3 discusses the possible factors of DES by comparing the Cox proportional hazard regression. These diseases are comorbidities of the patient as recorded in the patient’s National Health Insurance database. These comorbidities are pre-existent and have already been diagnosed prior to the orbital fracture. 

  1. Did the authors find increased of other eye diseases after orbital fracture? It would be important to determine the correlation between orbital fracture and the risk of other eye diseases.

Re: In Table 2, hordeleum,  blepharitis, and glaucoma were studied, of which hordeleum and glaucoma were significant. These three conditions were studied due to the fact that these are the conditions most likely to lead to dry eye syndrome. However, we did not find an increase in any of these conditions after orbital fracture. As a matter of fact, a decrease in hordeleum and glaucoma incidences were found after having experienced orbital fracture. 

  1. On pg 15. Please indicate which figure is referring these results. Please include only the data with the factors that have significant impact.

Re: The results on page 15 is a part of table 3. As these data are independent from each other, we felt it would be best represented in the form of a table. However, the factors that have significant impact have been pointed out in the result as your recommendation so readers can understand more easily. The following sentence was added, “Significant factors of dry eye syndrome include orbital fracture, female, 18- to 29-year age group, multiple sclerosis, diffuse diseases of connective tissue, blepharitis, hordeolum, glaucoma and Bell's palsy ”

  1. Please elaborate the importance of this findings in the discussion section. What kind of preventive therapy or therapeutic treatment can be considered?

Re: This study confirms the hypothesis that orbital fracture will indeed lead to a higher incidence of dry eyes. Therefore, we recommend that patients who have been diagnosed with orbital fractures should be routinely checked for dry eye symptoms so that earlier treatment can be initiated if confirmed. 

  1. The limitation of this study mentioned at the end of the discussion section should be mentioned at the beginning or in the method section prior to the interpretation of the results.

Re: Thank you for your suggestion. However, to our experience, study limitations have always been discussed in the latter part of the discussion section. The limitations are the concerns for bias, not the method as to how the study was carried out. Therefore, it seems more suitable to keep the limitation at the current discussion section. 

  1. Overall, English grammar should be revised. Eg. Section 2.4 “having received a diagnosis of DES diagnosis”

Re: Thank you for pointing this out. It has been corrected to “having received a diagnosis of DES”

  1. Check the fonts. Eg second paragraph on the introduction and third paragraph on the discussion section.

Re: Thank you for pointing this out. This has been corrected.

Reviewer 2 Report

Hsu et al. report the results of a large-scale retrospective investigation of whether orbital blow-out fractures are associated with dry eye.  While this could be assumed to already be in evidence (see Boyette et al., Clin Ophthalmol. 2015; 9: 2127–2137; Pansell et al., Craniomaxillofac Trauma Reconstr. 2012 Mar; 5(1): 1–6 for two examples), but this manuscript is unique in that it is a nationwide study in Taiwan and employs a comprehensive, multi-factorial analysis of dry eyes and blowout fractures. My low evaluation of the manuscript's "Presentation" is due to needed improvements in English and overall formatting.

Suggestions to improve the manuscript:

  1. Minor: I suggest including the total number of controls in the sentence "...group was age- and gender-matched to four individuals without orbital fracture..."
  2. Minor: "No matter..." at bottom of Pg 1 should be changed to "Regardless of whether..."
  3. Minor: "Base on clinical experience, we have hypothesized that orbital fracture..."  should be "Based on..."
  4. Minor: There is a formatting error in that same paragraph. "Little evidence..." is a larger font. (Same occurs on Pg 24.)
  5. MAJOR: The statement "little evidence" may be too strong. I believe there are sound investigations into this topic, and I believe them to be academic.
  6. Minor: There is a formatting issue with the whole paper. It's left margins are too wide, but that should be fixed with the typesetting staff.
  7.  Minor: Bottom of Pg. 2-"In dry eye patients, decrease of visual acuity associated with daily acts of gazing had been proved[12]" This is an awkward sentence.
  8. MAJOR:  The paper is relatively well-written, but it would greatly benefit from an English language service.
  9. MAJOR: Perhaps I missed it, but what is meant by "105 person years" in this sentence on Pg 5? "The incidence rate (per 105 person-years) of..."

Author Response

  1. Minor: I suggest including the total number of controls in the sentence "...group was age- and gender-matched to four individuals without orbital fracture..."

RE: Thank you for your suggestion. We have changed the abstract to include the total number of study and control groups in this cohort. 

“We studied a cohort from the Taiwan National Health Insurance da-tabase (NHIRD). A total of 46,179 and 184,716 participants were enrolled in the study and control groups, respectively. Then each patient in the case group was age- and gender-matched to four individuals without orbital fracture that served as the control group.”

  1. Minor: "No matter..." at bottom of Pg 1 should be changed to "Regardless of whether..."

RE: Thank you for pointing this out. This has been corrected. 

  1. Minor: "Base on clinical experience, we have hypothesized that orbital fracture..."  should be "Based on..."

RE: Thank you for pointing this out. This has been corrected. 

  1. Minor: There is a formatting error in that same paragraph. "Little evidence..." is a larger font. (Same occurs on Pg 24.)

RE: Thank you for pointing this out. It has been corrected. 

  1. MAJOR: The statement "little evidence" may be too strong. I believe there are sound investigations into this topic, and I believe them to be academic.

RE: Thank you for your recommendation. We have changed the statement to “There has yet to be any large population study to support this hypothesis. Therefore, we conducted a longitudinal nationwide population-based cohort study via the use of the Taiwan National Health Insurance Research Database (NHIRD).” 

  1. Minor: There is a formatting issue with the whole paper. It's left margins are too wide, but that should be fixed with the typesetting staff.

RE:Thank you for pointing this out. It has been corrected. 

  1. Minor: Bottom of Pg. 2-"In dry eye patients, decrease of visual acuity associated with daily acts of gazing had been proved[12]" This is an awkward sentence.

RE: Thank you for your recommendation. We have changed the statement to “A decrease in visual acuity associated with daily acts of gazing has been proved in dry eye patients. ”

  1. MAJOR:  The paper is relatively well-written, but it would greatly benefit from an English language service.

RE:Thank you for your suggestion. We have asked an English language service to revise this paper. 

  1. MAJOR: Perhaps I missed it, but what is meant by "105 person years" in this sentence on Pg 5? "The incidence rate (per 105 person-years) of..."

RE: The definition of person years is the type of measurement taken into account both the number of people in the study and the amount of time each person spends in the study. For example, a study that followed 1000 people for 1 year would contain 1000 person years of data.

Reviewer 3 Report

Thank you very much for submitting this very interesting paper! Please see the following recommendations:

Please add a space before the in-text citation. There are different font sizes used, please make sure that the document has consistent formatting.

Hypothesis:  Please add anatomical/physiological rationale to your hypothesis.

The paragraph about the dry eye is not related to the topic of the paper. For  the reader, it would be helpful if there would be a clearer description of dry eye and how ocular dryness could be related to ocular trauma.

Methods (2.1): Please add more detail about the random sampling.

Very nice figure 1!

Table 1: please provide abbreviations below or better avoid using abbreviation in the table.

Figure  2: Y-axis should go up to 1.

Table 3 should be supplementary.

This is a very interesting and relevant study. However, all tables are very hard to read and the layout of the paper makes it hard to access the information.   The authors should consider rearranging the data/ structure of the paper to make it more accessible. 

Author Response

  1. Please add a space before the in-text citation. There are different font sizes used, please make sure that the document has consistent formatting.

RE: Thank you for the heads up. We apologize for the font size problem. The document has been edited and double checked in order to make sure the format is consistent throughout. 

  1. Hypothesis:  Please add anatomical/physiological rationale to your hypothesis.

RE: The anatomical rationale of the hypothesis has been discussed in the third paragraph of the discussion, while the physiological rationale has been discussed in the fifth paragraphs of the discussion. 

  1. The paragraph about the dry eye is not related to the topic of the paper. For the reader, it would be helpful if there would be a clearer description of dry eye and how ocular dryness could be related to ocular trauma.

RE: Rather than trying to prove dry eyes being related to ocular trauma, we are actually trying to correlate whether ocular trauma would lead to dry eyes. Ocular trauma might lead to a decrease in tear gland secretion, which would then result in dry eye syndrome. The possible mechanisms are discussed in paragraph 3 of the discussion. 

  1. Methods (2.1): Please add more detail about the random sampling.

RE: Longitudinal Health Insurance Database (LHID) was a subset of the NHIRD. It contained information from 2 million people, and was used in the present study that randomly sampled individuals between 2000 and 2015. There was no significant difference in the distribution of sex, age, and insured premium between the LHID and the original NHIRD.

  1. Very nice figure 1!

RE: Thank you!

  1. Table 1: please provide abbreviations below or better avoid using abbreviation in the table.

RE: Abbreviations are added to table 1 (using tracking changes).

  1. Figure  2: Y-axis should go up to 1.

RE: We agree that in normal circumstances, the y-axis in a Kaplan-Meier curve should go up to 1.  However, due to the small incidence of cumulative risks, setting the Y-axis at 1 will cause the graph to be hard to interpret for the readers. Therefore, we set the y-axis to 0.01.

  1. Table 3 should be supplementary.

RE: Table 3 is modified. However, we still hope to present this table in the main content. 

  1. This is a very interesting and relevant study. However, all tables are very hard to read and the layout of the paper makes it hard to access the information. The authors should consider rearranging the data/ structure of the paper to make it more accessible.

RE: We are glad to know that you find this topic to be interesting and relevant. The tables have been modified in order to make it easier to read. Hopefully, after the revision, the readers can find the data to be more accessible.   

Reviewer 4 Report

However, this manusript contains data that would be of interest for our readers, it must be improved because the methodology of the study is flawn. Dry eye disease is a complex syndrome and it can not be definied only with ICD-9 code 375-15 also the prevalence of DES in general population of Taiwan is much higher than 0.11 %.

Author Response

RE: Thank you for your input. However, there are several things we should clarify. First of all, the 0.11% comes from the control group; this is the incident rate, not the prevalence of DES. We agree that the prevalence rate in Taiwan for DES should be much higher, as the majority of the DES population are the elderly, but the control group is based on case group age- and gender-matched groups. However, orbital fractures occur more often in the young, so the age group of the control groups will be younger as well. Therefore, the incident rate of DES in these groups should be lower, which is why we cannot compare it to that of the general population prevalence rate. 

Round 2

Reviewer 4 Report

According to the Definition and Classification Subcommittee of the International Dry Eye Workshop from year 2007 "Dry eye disease (DED) is a multifactorial disease of the tears and the ocular surface that results in symptoms of discomfort, visual disturbance, and tear film instability with potential damage to the ocular surface. It is accompanied by increased osmolarity of the tear film and the inflammation of the ocular surface" There is no definition of dry eye in the manuscript as well as what kind of dignostic tests were used to establish the diagnosis. The dry eye is very common in young people due to massive use of mobile electronic devices as well as computers so the study methodology is flawn and it must be improved before further consideration of publication process. 

Author Response

Dear Reviewer,

Thank you for the valuable suggestions! Our reply is as below.

Question: According to the Definition and Classification Subcommittee of the International Dry Eye Workshop from year 2007 “Dry eye disease (DED) is a multifactorial disease of the tears and the ocular surface that results in symptoms of discomfort, visual disturbance, and tear film instability with potential damage to the ocular surface. It is accompanied by increased osmolarity of the tear film and the inflammation of the ocular surface”. There is no definition of dry eye in the manuscript as well as what kind of diagnostic tests were used to establish the diagnosis.

        Thank you for pointing out that we did not give a definition for dry eye in the manuscript. We have now added the definition that you have kindly provided to the manuscript.

This study is based on Taiwan National Health Insurance database (NHIRD). Therefore, the diagnostic methods are those as performed clinically by the board-certified ophthalmologists in Taiwan.

According to the paper “Diagnosis of Dry Eye Disease and Emerging Technologies” published in Clin Ophthalmol. 2014, the current methods for the diagnosis of DES includes a slit-lamp examination with and without different stains, including fluorescein, rose bengal, and lissamine green. Other methods are the Schirmer test, tear function index, tear break-up time, and functional visual acuity.  Similarly, in the paper that you have quoted “The definition and classification of dry eye disease: report of the Definition and Classification Subcommittee of the International Dry Eye WorkShop (2007). 2007;5(2):75-92.", visual or discomfort symptoms along with signs such as conjunctival staining, conjunctival injection, corneal staining, corneal/tear signs, MGD, TFBUT, and Shirmer’s score help to diagnose dry eyes.  In Taiwan, clinical diagnosis by board-certified ophthalmologists follows the same methods as the two articles mentioned above. In addition, in order for the diagnose of dry eye to be included in Taiwan National Health Insurance database (NHIRD), diagnostic tests along with their results must be attached in the patient file. Therefore, the use of NHIRD diagnosis for dry eyes is credible even for research purposes.

Question: The dry eye is very common in young people due to massive use of mobile electronic devices as well as computers so the study methodology is flawed and it must be improved before further consideration of publication process.

        We agree that the use of electronic devices causes the prevalence rate in the young population to rise. However, in this study, the case group (with orbital fracture) and the control group (without orbital fracture) were age- and gender-matched. Therefore, both groups have a similar number of young population within the groups. As a result, this should negate the difference in incident rate of dry eyes as caused by the usage of electronic technologies for the two populations. Though the use of electronic devices leading to dry eyes might result in a higher than intended incident rate and prevalence rate, the inter-group comparison of dry eyes with and without ocular trauma should still be comparable.

        However, Taiwan National Health Insurance database (NHIRD) does not record how frequently the electronic devices are used. This has been previously pointed out in the limitation section. We will provide further emphasis on the manuscript to show that video terminal display exposure includes the use of mobile and electronic devices.

This manuscript is a resubmission of an earlier submission. The following is a list of the peer review reports and author responses from that submission.